# Identification Fluidity Method to Determine Suitability of Weathered and River Sand for Constructions Purposes

**Haoyu Zuo** [1,*], **Jin Li** [1,*], **Li Zhu** [2], **Degang Cheng** [2] and **De Chang** [2]

1    School of Transportation Civil Engineering, Shandong Jiaotong University, Jinan 250357, China
2    Jinan Kingyue Highway Engineering Company Ltd., Jinan 250409, China
*    Correspondence: 21107008@stu.sdjtu.edu.cn (H.Z.); sdzblijin@163.com (J.L.); Tel.: +86-18678092667 (H.Z.); +86-13678824225 (J.L.)

**Abstract:** At present, in order to comply with the development of the "the Belt and Road Initiatives", the country is accelerating the pace of construction and increasing the demand for construction river sand. However, the quality of construction river sand is uncontrollable, and its shape is very similar to that of weathered sand. Therefore, using inferior weathered sand and mixed sand as inferior substitute sand in the market is prohibited, resulting in an increase in the difficulty coefficient of quality control of concrete fine aggregate in actual projects. This lays hidden dangers for the construction quality of the project. It is urgent to improve the quality control, testing, and detection process of river sand. Due to the long-term weathering of weathered sand, its density is small, and there are many pores, which leads to the material's water absorption rate is higher than that of standard sand and river sand during fluidity tests. This paper takes this as a breakthrough point, reveals the variation law of fluidity loss under different variables, and explores a method for effectively screening low-quality sand and gravel. Through the silt content test (screening and washing method), the low-quality sand is preliminarily screened out, the mortar ratio is designed, and the fluidity test is carried out to compare the difference in fluidity loss of different types of mortar; determine the loss threshold range (mobility loss ≤ 15 mm) according to the mobility test results of the control group, and determine the qualification standard by comparing the measured mobility loss of the unknown sample with the loss threshold range. When the mobility loss is within the loss threshold range, the sample is qualified river sand. Otherwise, it is weathered sand or chowder sand. This method establishes a complete detection scheme for distinguishing weathered sand and river sand through mud content tests and mobility loss tests, solves the difficult problem of river sand quality control in engineering applications, and effectively eliminates the phenomenon of using low-quality weathered sand as river sand in the sand and gravel material market, thus avoiding congenital defects in concrete homogeneity.

**Keywords:** river sand; weathered sand; mud content; fluidity preface





## 1. Introduction

In evaluating the quality of engineering construction in civil engineering, similarly to other types of construction, it is necessary to apply a systemic approach. The quality in the previous international standard ISO 8402 was defined as the sum of characteristics of the product or service that reflect their ability to meet the stated and implied needs of the customers. The product or the construction product of a building process is the engineering construction representative of the highly expensive product from a majority of the range of works [1]. The standard EN ISO 9000 [2] defines quality as the degree to which a set of inherent characteristics fulfills requirements. Pavement roads are required to be designed, built, maintained, and disposed of at a reasonable price, with reasonable quality, respecting the relevant requirements of users and their surrounding residents and the principles of sustainable development during the life cycle.

Construction river sand is an indispensable part of the fine aggregates in pavement road engineering construction. It refers to the construction materials with certain quality standards formed by the action of natural forces, such as the impact and erosion of river water. After drying and screening, it can be widely used in various dry mortars and plays an irreplaceable role in the construction industry. Weathered sand is a kind of material that is broken and loose after a long exposure to solar radiation, the atmosphere, and water. Its durability is poorer than ordinary soil materials, its strength is weaker, its physical and mechanical properties are unstable, and it contains a certain amount of fine soil particles.

Compared with river sand, weathered sand is cheap and easy to obtain, but its various property indicators cannot meet the requirements of construction materials. If it is used for concrete mixing, it will have a huge impact on the quality of concrete, causing rapid loss of concrete slump and affecting its strength, durability, and workability. Due to improper use of weathered sand, the concrete strength is seriously lower than the design strength grade, and the phenomenon of concrete cracking frequently occurs, which brings some difficulties to construction quality control.

Generally, the surface of river sand is smooth, the particles are smooth and relatively clean, and the strength is high, so it is not easy to twist with fingers. However, the weathered sand particles are angular, with a prickly feeling when rubbed by hand. The size distribution is uneven and contains a large number of fine soil particles. At present, the common discrimination method in the industry is to distinguish by hand and eye. If only judged by experience, the distinction between the two is vulnerable to subjective factors, and the discrimination error is large, leading to the increase in the difficulty coefficient of raw material quality control in engineering applications. As the quality of inland river sand in Shandong Province is relatively poor and is mixed with weathered sand, the durability of weathered sand is poorer than that of ordinary soil materials, and its physical and mechanical properties are more unstable, it is easy to lay hidden dangers for project construction quality.

Yan Zhenqiang [3], from the Shandong Jianzhu University, systematically studied the road performance and microscopic characteristics of cement-stabilized weathered sand. First, based on mastering the basic physical and mechanical properties of weathered sand, a series of indoor tests were carried out to determine the influence of cement content and curing age on the mechanical properties of weathered sand. Through the dry shrinkage test, dry wet cycle test, and freeze-thaw cycle test, the influence of environmental changes on the durability of cement-stabilized weathered sand is analyzed. In addition, the microstructure characteristics of cement-stabilized weathered sand under different working conditions are analyzed through a scanning electron microscope test. Based on this, the strong growth mechanism of cement-stabilized weathered sand and the deterioration mechanism under dry, wet, and freeze-thaw cycles are discussed.

Lin Yunken [4] of the Yongjia Rongchang Concrete Co., Ltd. (Wenzhou, China) studied the influence of river sand silt content on the performance of machine-made sand concrete and tested the amount of a mixture, working performance and compressive strength of concrete mixed with river sand with different silt content. It is found that river sand containing mud will increase the absorption of admixtures, reduce the working performance of concrete, reduce the slump retaining capacity of concrete, and thus reduce the engineering properties of concrete.

Zhao Wenkun [5] of the China Communications Construction Company First Harbor Engineering Co., Ltd. (Beijing, China) studied the difference in the microstructure of manufactured sand in Shiling Quarry and river sand in Qingping Quarry through Nikon SMZ800N body microscope and computer graphics processing technology and then analyzed its impact on the mechanical properties of concrete from the difference in roundness coefficient and particle morphology.

Zhang Xiao [6] of the Liaoning Provincial Communications Planning and Design Institute Co., Ltd. (Shenyang, China) found that with the increase in river sand content, the mechanical properties of UHPC first increased and then decreased, and the working perfor-

mance gradually improved. When river sand content was 30%, the compressive strength, and flexural strength reached the maximum, the tensile strength of UHPC continued to decrease, and the durability of UHPC was better due to the improvement of chloride ion penetration resistance. Domestic and foreign scholars have conducted a lot of research on the microscopic characteristics of weathered sand and river sand and the improvement and application of machine-made sand and river sand in concrete [7–26].

Through a series of tests, this paper compares the differences in physical properties between weathered sand and river sand and then puts forward a method to judge the quality of river sand. It uses scientific and reasonable means to evaluate the quality of sand and gravel, providing scientific standards and norms for practical engineering applications. The river sand and unknown sand samples required for the test in this paper are taken from the beach along the Yellow River in Qihe River, Dezhou, Shandong Province, and the weathered sand samples are taken from the weathered rock in Wulian, Rizhao, Shandong Province. In the application process, this method can reflect the problem of water absorption of materials to reflect the workability and fluidity of the tested materials used for mixing concrete. The test equipment and methods used are the existing mud content and fluidity test equipment and methods. The process is simple, convenient, fast, saves manpower and material resources, and is suitable for standardized management with high accuracy. It avoids the subjective factors affecting the results only based on subjective experience. The identification of defects with large errors reduces the difficulty of raw material quality control in engineering applications, ensures the safety of engineering construction, and can create more economic value and social benefits.

## 2. Basic Physical Properties Test of River Sand and Weathered Sand

### 2.1. Natural Water Content Test

Sample the soil samples and place them in a cool place for sealed storage and drying. Take a clean and dry aluminum box and say its mass is $m_1$. Put the soil sample in the sealed bag into the aluminum box, close the lid and call its mass $m_2$. Set the temperature of the oven to 105 °C; when the oven reaches the set temperature, take off the lid of the aluminum box in the oven for drying. After drying, take out the aluminum box and quickly cover the box cover. After the sample cools, weigh the aluminum box and the dried sample mass $m_3$. When two consecutive weighing differences are unchanged, drying ends. The natural water content test results are shown in Table 1.

**Table 1.** Natural water content test.

| Aluminum Box Number | 1 | 2 | 3 | 4 |
|:---:|:---:|:---:|:---:|:---:|
| Aluminum box mass/g | 88.6 | 102.6 | 88.6 | 102.6 |
| Aluminum box + wet soil total mass/g | 160.3 | 155.9 | 123.6 | 143.4 |
| Aluminum box + dry soil total mass/g | 163.4 | 159.1 | 127.5 | 147.5 |
| Water content/% | 3.1 | 3.2 | 3.9 | 4.1 |
| Average moisture content/% | | | 3.6 | |

Calculate the natural moisture content of weathered sand:

$$\omega = \frac{m_2 - m_3}{m_3 - m_1} \times 100\% \tag{1}$$

$\omega$—moisture content (%); $m_1$—Mass of aluminum box (g); $m_2$—Total mass of the aluminum box and wet soil (g); $m_3$—Total mass of the aluminum box and dry soil (g)

Test results:

According to the above method, the average water content of weathered sand measured is 3.6%. Similarly, the average water content of the river sand sample is 5.8%. The average water content of river sand is greater than that of weathered sand. However, the

water content will produce large errors due to environmental impact during the transportation and test of sand and gravel materials, so it cannot be used as a basis for distinguishing weathered sand from river sand.

*2.2. Particle Analysis Test*

(1)    Test method

Representative samples of dry soil were taken out by the quartering method, and 2 mm samples were screened in batches. Samples larger than 2 mm are passed through each layer of the coarse sieve in turn, and the remaining soil samples on the sieve are weighed separately. Shake the soil sample under the 2 mm sieve through the vibrating sieve machine for 10 min, then start with the screen with the largest aperture and gently pat and shake at the bottom of the sieve where the white paper is placed until the mass under the sieve does not exceed 1% of the remaining mass of the sieve level per minute. Place all leaking soil particles in the next level sieve, brush the soil sample on the sieve with a soft brush and weigh separately.

(2)    Test results and analysis

The test results of granular analysis of weathered sand and river sand are shown in Table 2, and the particle distribution curve is shown in Figure 1.

**Table 2.** Particle analysis results.

| Screen Size/mm | 40 | 20 | 10 | 5 | 2 | 1 | 0.5 | 0.25 | 0.075 |
|---|---|---|---|---|---|---|---|---|---|
| Aeolian sand passing mass percentage/% | 100 | 96.1 | 90.2 | 85.2 | 73.8 | 63.7 | 37.6 | 18.6 | 2.7 |
| River sand passing mass percentage/% | 100 | 100 | 96.4 | 89.7 | 79.4 | 49.3 | 19.1 | 5.5 | 2.2 |

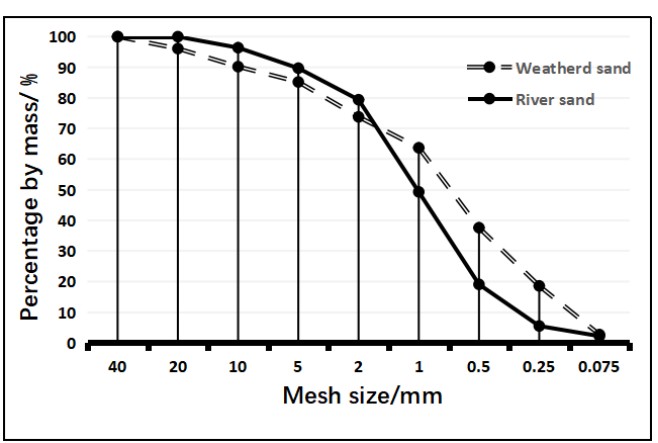

**Figure 1.** Particle analysis result plot.

From Table 2 and Figure 1, it can be concluded that the content of the weathered gravel grain group of this group of samples is 26.2%, the content of the sand grain group is 73.8%, the content of the fine grain group is less than 5%, the non-uniformity coefficient Cu is 5.6, and the curvature coefficient Cc is 1.1, which are well-graded sands. The content of the river gravel group is 21.6%, the content of the sand group is 78.4%, the content of the fine group is less than 5%, the non-uniformity coefficient Cu is 10.7, and the curvature coefficient Cc is 3.7. Therefore, the size distribution of river sand is more concentrated, and the size distribution of weathered sand is more dispersed.

**3. Apparent Density and Bulk Density Test of Sand**

*3.1. Test Method for Apparent Density of Sand*

(1)    Weigh 300 g ($G_0$) of dried river sand and weathered sand samples, respectively;

(2)  Fill the volumetric flask with water to the bottleneck scale line, wipe off the water outside the flask, and weigh its mass ($G_2$);

(3)  Pour out the water from the inside of the volumetric flask and the rest to about 1/3 of the height of the ball. Add 300 g of sand into the volumetric flask, tilt the volumetric flask at an angle of about 45 degrees, and make the sample stir fully in the water (remove bubbles). After standing for a period of time, add water to the bottleneck scale line with a dropper, wipe off the water outside the bottle, and weigh its mass ($G_1$).

The apparent density test of sand is shown in Figure 2.

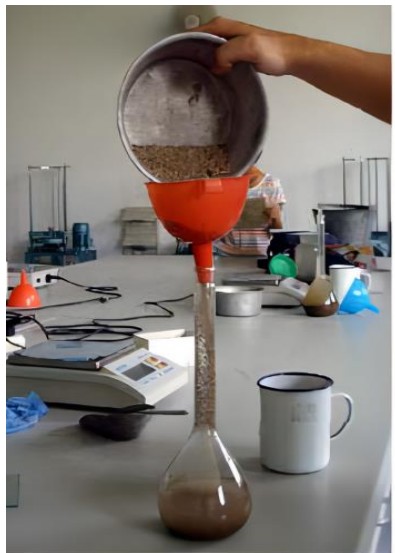

**Figure 2.** Sand apparent density test.

*3.2. Data Processing*

The apparent density of sand is calculated as follows: weathered sand is $\rho_1$, river sand is $\rho_2$, and the result is accurate to 10 kg/m$^3$. The correction coefficient of the effect of different water temperatures on the apparent density of sand is shown in Table 3.

$$\rho_0 = \left( \frac{G_0}{G_0 + G_2 - G_1} - \alpha_t \right) \times \rho_{water} \tag{2}$$

**Table 3.** Correction coefficient of the influence of different water temperatures.

| Water Temperature/°C | 15 | 16 | 17 | 18 | 19 | 20 | 21 | 22 | 23 | 24 | 25 |
|---|---|---|---|---|---|---|---|---|---|---|---|
| $\alpha_t$ | 0.002 | 0.003 | 0.003 | 0.004 | 0.004 | 0.005 | 0.005 | 0.006 | 0.006 | 0.007 | 0.008 |

$\alpha_t$: Correction coefficient of the influence of different water temperatures on the apparent density of sand.

$\rho_{water}$ = 997 Kg/m$^3$, Water temperature = 18 °C.

$$\rho_1 = \left( \frac{300}{300 + 400 - 584.6} - 0.04 \right) \times 997 = 2551.97 \tag{3}$$

$$\rho_2 = \left( \frac{300}{300 + 400 - 595.3} - 0.04 \right) \times 997 = 2816.85 \tag{4}$$

Take the arithmetic mean of the two parallel sample test results as the final result. The difference between the measurement results should be less than 20 Kg/m$^3$. Otherwise, it

should be reperformed. According to the formula, the apparent density of weathered sand is 2573, and that of river sand is 2810. The apparent density of river sand is slightly higher than that of weathered sand.

*3.3. Test Method for Loose Bulk Density of Sand*

(1) Weigh the mass ($G_1$) of the standard container and measure the volume (V) of the standard container. Place the standard container under the blanking hopper to make the hopper align with the center;

(2) Load the sample into the blanking funnel, open the movable door, and let the sample slowly fall into the standard container until it is full and exceeds the opening of the standard container, then remove the funnel;

The loose bulk density test of sand is shown in Figures 3 and 4.

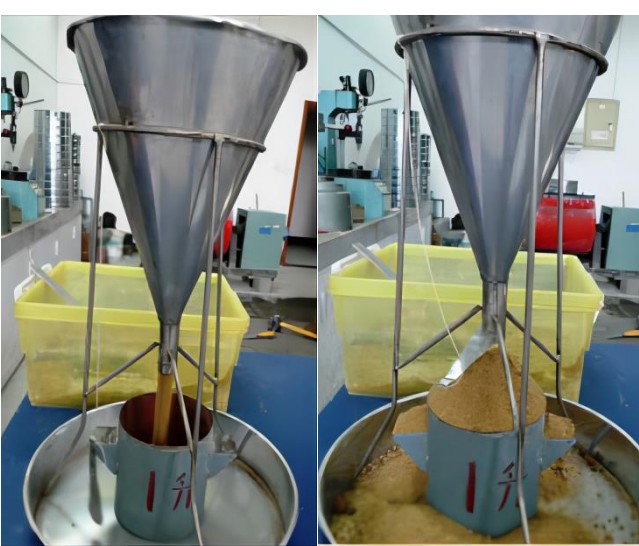

**Figure 3.** Feeding funnel charging.

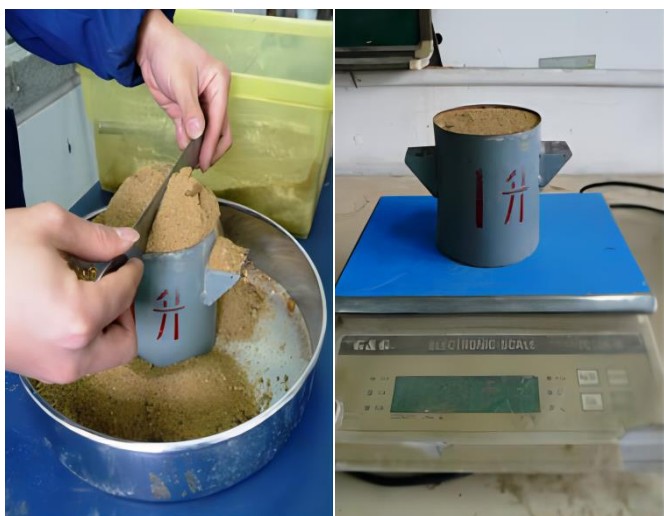

**Figure 4.** Scrape flat.

(3) Use a ruler to scrape the excess sample in the opposite direction along the center line of the barrel mouth and weigh its mass ($G_2$).

Data processing:

(1) Calculate the bulk density of the sample according to the following formula, accurate to 10 Kg/m$^3$.

$$\rho_0 = \frac{G_1 - G_2}{V} \tag{5}$$

(2)　Take the arithmetic mean of the test results of two parallel samples as the final result, accurate to 10 Kg/m$^3$.

$$\rho_1 = \frac{G_1 - G_2}{V} = 1576.3 \tag{6}$$

$$\rho_2 = \frac{G_1 - G_2}{V} = 1449.5 \tag{7}$$

According to the formula, the loose bulk density of river sand is 1560, and that of weathered sand is 1450.

To sum up, the apparent density and bulk density of river sand is slightly higher than that of weathered sand. It is speculated that the shape of weathered sand particles is irregular, and the void ratio is higher than that of river sand due to long-term weathering and erosion. However, since the difference in data comparison is not obvious, it cannot be used as a basis for distinguishing weathered sand from river sand.

## 4. Comparative Test on Fluidity Loss of Different Sand and Gravel Materials

By selecting standard sand, qualified river sand, qualified weathered sand with known silt content, and river sand materials with high silt content, the fluidity test is conducted under the condition of a standard temperature of the laboratory 20 ± 2 °C and relative humidity > 50%. The test results are shown in Table 4. Several groups of repeated tests on fluidity loss of river sand and weathered sand under different mud content conditions are carried out, and the results are shown in Figures 5 and 6:

**Table 4.** Fluidity loss table.

| Group | Standard Sand | Qualified River Sand 1 | Qualified River Sand 2 | Weathered Sand 1 | Weathered Sand 2 | Silt 3.6% River Sand | Mud-3.2% River Sand |
|---|---|---|---|---|---|---|---|
| First diameter value (mm) | 300 | 300 | 297 | 281 | 274 | 262 | 270 |
| Stand for 30 min diameter value (mm) | 300 | 300 | 295 | 212 | 230 | 196 | 211 |
| Fluidity loss s (mm) | 0 | 0 | 2 | 69 | 44 | 66 | 59 |

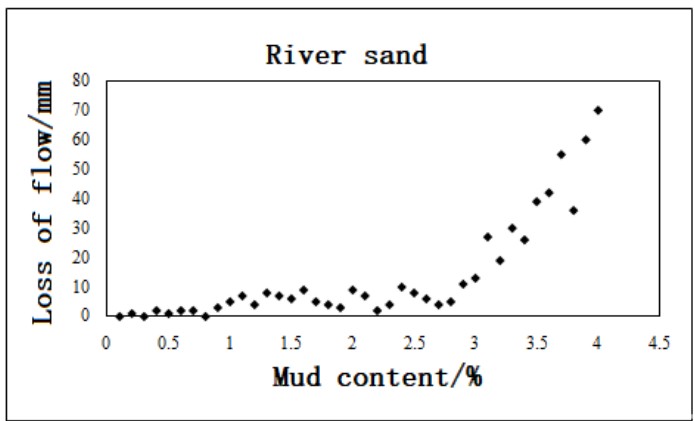

**Figure 5.** Flow loss of river sand with different silt content.

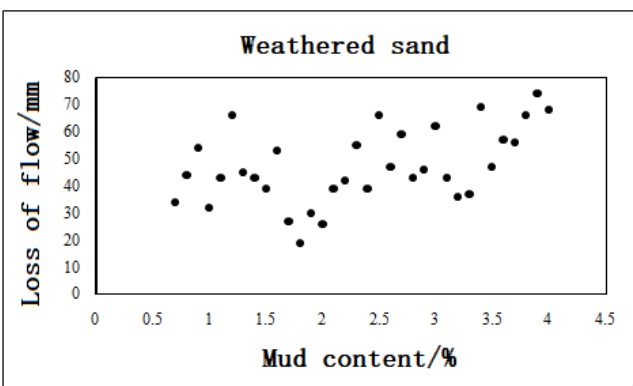

**Figure 6.** Flowability loss of weathered sand with different mud content.

According to test data, the results show that:

(1) The loss value of fluidity of river sand with unqualified silt content is large. In combination with Figure 5, the relationship between the loss of fluidity of river sand and silt content, the loss value of fluidity of river sand with silt content of more than 3% changes greatly. When the silt content is less than 3%, the loss of fluidity of river sand is less than 15 mm. Therefore, it can be seen that the silt content has a great impact on mobility. First, it is necessary to exclude the impact of the silt content on the material fluidity loss test.

(2) The fluidity loss value of weathered sand is significantly larger than that of standard sand and qualified river sand. In combination with Figure 6, the relationship between fluidity loss and mud content of weathered sand, if the mud content of mobilized materials meets the standard, and the fluidity loss value is >15 mm, it can be judged as weathered sand, that is, the river sand is unqualified.

(3) The fluidity loss of qualified river sand is less than 15 mm, and the fluidity loss of weathered sand or unqualified sand is more than 15 mm, so the threshold range of qualified material fluidity loss is determined: under the condition of qualified mud content when the fluidity loss is more than 15 mm, it can be judged as weathered sand or mixed sand. The smaller the fluidity loss, the better the river sand quality.

## 5. Quality Inspection Test of Newly Arrived Sand and Gravel

Then, the quality of unknown sand and gravel is detected through two embodiments of mud content and a fluidity test.

### 5.1. Implementation I

(1) Mud content test

The new materials shall be tested in accordance with the requirements of T 0333-2000 fine aggregate mud content test (sieve washing method) in JTG E42-2005 Test Rules for Aggregates of Highway Engineering [27]. The mud content of the samples shall be calculated to 0.1% according to the formula.

$$Q_n = \frac{m' - m}{m} \times 100 \tag{8}$$

where: $Q_n$ is the mud content of the sample (%); $m$ is the weight of the dried sample before the test (g); $m'$ is the weight of the dried sample after the test (g).

The arithmetic mean of the test results of the two samples was used as the measurement value. When the difference between the two results exceeds 0.5%, a new sample should be taken for testing.

During the mud content test, samples of about 400 g (m) per mass are weighed according to the test procedures. The final mud content shall meet the following requirements:

$Q_n$ 3.0 or less. If the mud content of the incoming material meets the requirements of this standard, proceed to the next step of detection; Otherwise, if the batch of material is unqualified, the yard will not receive it.

For the three new batches of materials, 400 g is taken, respectively, for the mud content detection. The test results are shown in Table 5:

**Table 5.** Mud content test results of Embodiment 1.

| The Sample | Sample 1 | Sample 2 | Sample 3 |
|---|---|---|---|
| Mud content $Qn$ (%) | 3.8 | 1.9 | 3.3 |

Under the same test condition, the mud contents of sample 1 and sample 3 are 3.8% and 3.3%, respectively, which is not satisfied $Q_n$ (mud content) $\leq 3.0$ requirements. Sample 2 meets the mud content standard and can be tested for fluidity in the next step after preliminary screening. Sample 2 is selected as the next step of the test material, hereafter referred to as the unknown material.

(2) Fluidity test

The screened sample mortar is prepared from the sample whose mud content is not greater than the set threshold of mud content. The material ratio and mixing of the sample mortar, standard sand mortar, and river sand mortar are based on the requirements specified in GBT 17671-1999 Test Method for Strength of Cement Mortar (ISO Method) [28] and JC/T681-2005 Planetary Cement Mortar Mixer [29].

According to the requirements of GBT 17671-1999 Test Method for Strength of Cement Mortar (ISO Method), the quality mix proportion of cement mortar shall be one part of cement, three parts of sand, and half water, that is, 1:3:0.5 for cement: standard sand: water. In order to better simulate the actual effect of concrete, a 1% water-reducing agent shall be added. The final design mix ratio of cement: sand: water is 2.5:5:1. The specific preparation method is as follows: take water according to the set mass ratio scale and add cement into the pot. The water reducer makes its state select the appropriate adding time. If it is a powder water reducer, add it into the pot together with cement and other powders, and the liquid water reducer is added into the pot together with water. The cement used is ordinary Portland cement.

After using the cement mortar mixer to mix at a low speed for the set time, evenly add the sample into the mixture. The speed of mixing at low speed is: the rotation of the mixer shaft is $140 \pm 5$ r/min, and the revolution is $62 \pm 5$ r/min. According to the specification requirements, completely mix the mixture evenly and complete the preparation of the sample mortar. The setting time is 30 s.

The preparation method of standard sand mortar and river sand mortar is the same as that of sample mortar, except that the standard sand mortar is obtained by replacing the sample with ISO standard sand of the same quality, and the river sand mortar is obtained by replacing the sample with river sand of the same quality.

The standard sand group is set as control group 1, river sand 1, river sand 2, and river sand 3 corresponding to control group 2, control group 3, and control group 4, respectively, and the samples to be judged are test groups.

Control group 1: weigh 400 g cement, 800 g standard sand, 160 g water, and 4 g water reducer, respectively, and add water and cement into the pot in turn. The water reducer shall be added at the appropriate time according to its state: powder water reducer shall be added together with cement and other powders, and liquid water reducer shall be added together with water. After 30 s of low-speed mixing with a cement mortar mixer, add standard sand evenly at the beginning of the second 30 s. Mix the materials completely and evenly according to the specification requirements to complete the preparation of standard sand mortar.

In the same way as the control group 1, the control groups 2, 3, 4, and the test group (the standard sand is replaced with the corresponding river sand or sample of the same

gram) are mixed with cement mortar to complete the preparation of two groups of river sand mortar and sample mortar.

According to the test procedures of GBT-2419-2005 Method for Determining the Fluidity of Cement Mortar [30], under the condition that the standard temperature of the laboratory is $20 \pm 2$ °C and the relative humidity is >50%, take the materials separately, and carry out the fluidity test on five groups of newly mixed materials to determine their fluidity values. At this time, their fluidity values are recorded as d. At the same time, stand the mixed mortar for 30 min and measure its fluidity value, which is recorded as d′.

It shall be completed within 6 min from the time of adding water to the mortar to the time of measuring the diffusion diameter. After jumping the table, use a caliper to measure the diameters of the two directions perpendicular to each other on the bottom surface of the mortar, calculate the average value, and take an integer (mm). The average value is the cement mortar fluidity of the water volume.

Calculate the fluidity loss s of each group, and according to the test results of the four control groups, define the fluidity range of qualified sand materials, that is, the loss threshold range: if the fluidity value of the experimental group conforms to the range, the material is available river sand. Otherwise, the material does not meet the material requirements of the stockyard and is unqualified, which is weathered sand or chowder sand (river sand mixed with weathered sand).

Loss of fluidity $s = d - d'$

The fluidity test results are shown in Table 6.

**Table 6.** Fluidity test results of Embodiment 1.

| Group | Control Group 1 | Control Group 2 | Control Group 3 | Control Group 4 | Test Group |
|---|---|---|---|---|---|
| First diameter value (mm) | 300 | 288 | 295 | 300 | 275 |
| Stand for 30 min diameter value (mm) | 300 | 286 | 291 | 300 | 199 |
| Fluidity loss s (mm) | 0 | 2 | 4 | 0 | 76 |

Under this test scheme, the performance of control group 1 was good. When it jumped the table 23 times, it exceeded the 300 mm range of the disc table, and the first fluidity value was recorded as 300 mm. In control groups 2, 3, and 4, the initial mobility values were all between 280 mm–300 mm, and there was almost no change in the mobility values after standing for 30 min.

After standing for 30 min, compared with the control test, the material fluidity loss of the experimental group was serious, up to 76 mm, which could be determined that the material of the experimental group was blown sand or mixed sand.

According to the test results, the fluidity loss of problem materials is obviously larger, and the fluidity loss of qualified materials is less than 10 mm. The results of multiple tests can be combined to determine the range of qualified material mobility.

*5.2. Implementation II*

For the four new batches of materials, 400 g is taken, respectively, for the mud content detection. The test results are shown in Table 7:

**Table 7.** Mud content test results of Embodiment 2.

| The Sample | Sample 1 | Sample 2 | Sample 3 |
|---|---|---|---|
| Mud content $Qn$ (%) | 1.7 | 3.6 | 2.1 |



Among them, the mud content of sample 2 is 3.6%, which does not meet $Qn \leq 3$. The clay content of samples 1 and 3 met the requirements, and they were selected for the next step of the fluidity test.

Standard sand was taken and set as control group 1, river sand 1 and river sand 2 corresponding to control group 2 and control group 3, and sample 1 and sample 3 were set as unknown material 1 and unknown material 2 corresponding to test group 1 and test group 2.

Weigh 400 g cement, 800 g standard sand/river sand 1/river sand 2/unknown material 1/unknown material 2, 160 g water, and 4 g water reducing agent, respectively, and add water and cement into the pot in turn. The appropriate adding time of water reducing agent is selected according to its state: a powder water-reducing agent is added together with cement and other powder, and a liquid water-reducing agent is added together with water. After mixing with the cement mortar mixer at a low speed for 30 s, add the sand evenly at the beginning of the second 30 s, and mix the material completely and evenly according to the time required by the specification.

Five groups of fresh mix materials were tested for fluidity, and the fluidity values of fresh mix and material standing for 30 min were measured, respectively, to calculate the fluidity loss. The fluidity test results are shown in Table 8.

**Table 8.** Fluidity test results of Embodiment 2.

| Group | Control Group 1 | Control Group 2 | Control Group 3 | Test Group 1 | Test Group 2 |
|---|---|---|---|---|---|
| First diameter value (mm) | 300 | 287 | 291 | 300 | 285 |
| Stand for 30 min diameter value (mm) | 300 | 282 | 288 | 300 | 233 |
| Fluidity loss s (mm) | 0 | 5 | 3 | 0 | 52 |

Under the same test conditions, the fluidity of test group 1 is good, and the fluidity loss is 0, which is qualified river sand. The fluidity loss of test group 2 is serious, and its value is up to 52 mm. It can be judged as weathered sand or mixed sand, and the material is unqualified, so it will not be accepted.

The flow of qualified river sand can be determined from the above tests as shown in Figure 7.

This flow chart provides a method to judge the quality of river sand, including the following steps:

(1) Determine the mud content of sand and gravel with a fineness modulus of 2.3–3.0. If the measured mud content is greater than the set threshold of mud content (mud content $\leq 3.0$), the sample is unqualified. When measuring the mud content, carry out at least two mud content measurement tests, and take the arithmetic mean value as the mud content of the sample to be judged.

(2) The sample mortar is prepared from the sample whose mud content is not greater than the set threshold of mud content, and the fluidity of the sample mortar is tested to obtain the fluidity loss of the sample mortar between two test moments. The sample mortar is a mixture of cement, sample, and water with a set mass ratio, and the preparation method of the sample mortar is as follows: weigh the cement and water with a set mass ratio, add the sample after mixing for a set time, continue mixing for a set time, and complete the preparation of the sample mortar. When preparing the sample mortar, add a water reducer, which is mixed with cement and water. Water reducing agent is added in the preparation process of the standard sand mortar and river sand mortar.

(3) Set up multiple groups of control experiments to determine the range of loss threshold, and compare the obtained mobility loss with the range of loss threshold (mobility loss

value > 15 mm). When the mobility loss is within the range of the loss threshold, the sample is qualified as river sand. Otherwise, it is unqualified as weathered sand or mixed sand.

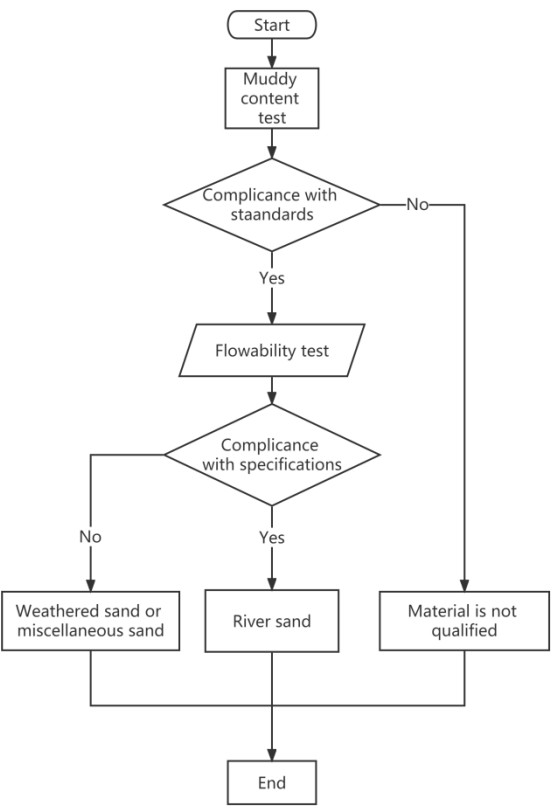

**Figure 7.** Flow chart of river sand inspection.

The mortar used for the fluidity test in the control experiment is a standard sand mortar and several groups of river sand mortar. The standard sand mortar is a mixture of cement, standard sand, and water with a set proportion, and the river sand mortar is a mixture of cement, river sand, and water with a set proportion.

The fluidity of sample mortar, standard sand mortar, and several groups of river sand mortar are tested by the mortar fluidity tester. The time interval between two test moments is 25 min–35 min.

In addition, the fluidity test is conducted at a temperature of $20 \pm 2\ °C$ and relative humidity of >50%.

## 6. Conclusions

(1) The average water content of river sand is slightly higher than that of weathered sand, and the water content varies greatly due to environmental factors in different regions, so it cannot be used as a basis for distinguishing weathered sand from river sand;

(2) In the particle analysis test, the non-uniformity coefficient and curvature coefficient show that the size distribution of river sand is more concentrated, and the size distribution of weathered sand is more dispersed. In the actual project, the mixing of different kinds of sand and gravel will affect the results of this parameter, so it cannot be used as a method to identify qualified sand;

(3) The apparent density and bulk density of river sand is slightly higher than that of weathered sand, but because the differences between data are not obvious, it cannot be used as the basis for distinguishing weathered sand from river sand;

(4) When the silt content is $\geq 3\%$, the fluidity loss value of river sand is relatively large. Therefore, the influence of silt content on the fluidity loss test of materials should be excluded before the fluidity test of mortar;

(5) The fluidity loss value of weathered sand is significantly larger than that of standard sand and qualified river sand. On the premise that the mud content of the mobilized material meets the standard, the fluidity loss value is >15 mm, which can be judged as weathered sand; that is, the river sand is unqualified;

(6) The fluidity loss of qualified river sand is less than 15 mm, and the fluidity loss of weathered sand or unqualified sand is more than 15 mm, so the threshold range of qualified material fluidity loss is determined: under the condition of qualified mud content when the fluidity loss is more than 15 mm, it can be judged as weathered sand or mixed sand. The smaller the fluidity loss, the better the river sand quality.

In order to improve the soil quality of the country and reduce the occurrence of natural disasters, Koki Nakao [31] of Japan conducted a visual and measurable assessment of the quality and performance of ground improvement through MPS-CAE analysis and worked through computer simulation. A series of operations performed by DRT and common (NT) RS-DMM are extracted using 3D models. Then, the internal condition of the ground and the displacement reduction performance are evaluated during each construction period. In the future, the distinction between weathered sand and river sand can be accurately calculated by similar numerical simulation methods to reduce judgment error. However, due to technical limitations, this method still has certain limitations. The method provided in this paper has a high reference value for the current construction field.

To summarize the above conclusions, this paper explores the method of reasonably distinguishing the weathered sand of river sand through several tests. Whether it is the test comparison of water content, particle size distribution, apparent density, and bulk density, the final difference between the two is almost insignificant and also confirms the similarity between river sand and weathered sand from a scientific point of view. Finally, two parameters, mud content, and fluidity, are selected to judge the quality of sand and stone samples. This method can control the quality of incoming river sand more accurately and make the inspection method standardized, which is convenient for strengthening laboratory management and suitable for popularization and application. In addition, during the application of this method, the water absorption of materials can be reflected, on the other hand, to reflect the workability and fluidity of the tested materials used for mixing concrete. It is suitable for standardized management and can be judged by the objective data obtained from the test with high accuracy. It avoids the defects that the results are affected by subjective factors, and the identification error is large only by the subjective experience judgment, which reduces the difficulty of raw material quality control in the engineering application. It has an important reference significance for the mobilization acceptance of sand and stone materials in the actual project.

**Author Contributions:** Conceptualization, H.Z.; methodology, H.Z. and J.L.; software, H.Z. and J.L.; resources, H.Z.; writing—original draft preparation, H.Z. and D.C. (Degang Cheng); writing—review and editing, H.Z. and D.C. (Degang Cheng); supervision, H.Z.; project administration, H.Z., L.Z. and D.C. (De Chang); funding acquisition, H.Z. and L.Z. All authors have read and agreed to the published version of the manuscript.

**Funding:** National Key Research and Development Program of China (2022YFB2601900).

**Institutional Review Board Statement:** Not applicable.

**Informed Consent Statement:** Not applicable.

**Data Availability Statement:** All data that support the findings of this study are included within the article.

**Conflicts of Interest:** The authors declare no conflict of interest.

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
