# Peer review of "Identification Fluidity Method to Determine Suitability of Weathered and River Sand for Constructions Purposes"

_coatings, doi:10.3390/coatings13020327_

Round 1
Reviewer 1 Report
Before making any recommendations for an interesting scientific article "Quality Identification Method Based on the Difference in Fluidity between Weathered Sand and River Sand", I would like to present the following generally statements on the reviewed article. In order for the article to be could published in the renowned journal Coatings, it needs significant improvements. For this purpose, I take the liberty of presenting the following mandatory requirements for inclusion and facultative requirements, the no inclusion of which I require to be credibly justified.
Mandatory requirements:
LNSA 10-28….Abstract… I would allow myself to take the following position regarding the abstract. In general, I expect from a high-quality scientific abstract a brief scientific summary of the solved problem, an explicit determination of scientific goals and corresponding methodology. From my point of view, I consider the abstract in the number of 19 lines to be above standard in scope and too focused on the solved research problem without a wider scientific context. With the validity of the first sentence "At present, there is no mature and systematic method to distinguish weathered sand from river sand and standardized specification in the industry, which increases the difficulty coefficient of quality control of concrete fine aggregate in actual projects, laying hidden dangers for project construction quality" I would allow myself to argue from a global point of view. I would dare to disagree with the authors' statement ...there is no mature and systematic method...even with regard to the presented number of references. It can be anticipated that construction technology companies have either directly or indirectly dealt with the given issue, but they consider it their know-how.
LNSA 31-100…1. Introduction… River sand refers to the building materials with certain quality standards formed in the river water by the action of natural forces, impact and erosion of the river water… There is no direct reference in the entire introduction, which is unacceptable for a scientific article. The solved research problem must result from a detailed studiesof world literature, its analysis, synthesis and relevant conclusions for the article in question. Personally, I like that the authors give priority to the use of locally suitable materials, which is part of my holistic premise of pavement structure design. However, the solved scientific problem needs to be solved in the context of a global approach. A similar problem of using conditionally suitable materials is described in "Design of road pavement using recycled aggregate".
LNSA 32-33… River sand refers to the building materials with certain quality standards formed in the river water by the action of natural forces, impact and erosion of the river water…. River sand refers to the building materials with certain quality standards formed in the river water by the action of natural forces, impact and erosion of the river water...River sand is a general term characterizing the origin and formation of sand, and not all sands are usable as building materials. I recommend the authors consider changing the term, for example, construction river sand or river sand suitable for construction purposes.
LNSA 191... 10Kg/m3...incorrect physical unit format.
LNSA 214… Figure 2. Flow loss of river sand with different silt content..incorrect number of the figure and the need to increase the graphic quality.
LNSA 265-268...The material ratio and mixing of the sample 265 mortar, standard sand mortar and river sand mortar are based on the requirements spec-266 ified in GBT 17671-1999 Test Method for Strength of Cement Mortar (ISO Method) and 267 JC/T681-2005 Planetary Cement Mortar Mixer...it is necessary to mention the presented standard methods in the references. It also applies to other parts of the article.
LNSA 364...Figure 4. Flow chart of river sand inspection... It is necessary to rework the image in order to significantly improve its graphic quality. It would also be appropriate to increase the expressive value of the image, to specify "finenesses modulus filtering" in more detail, to include directly in the figure or the related text references to "compliance with standard"...
LNSA 395-432…6. Conclusions… Conslusions need to be comprehensively revised. As I state in more detail in the requirements for revising the abstract and introduction, in the conclusions it is necessary to proceed from the general characteristics of the problem to specific results. The general characteristics of the described issue in the context of the most important works of foreign researches are completely absent. Personally, I consider it appropriate when the conclusions also include references to renowned related foreign publications, an explicitly stated authors contribution and an indication of possible continuation of research in the subject area.
LNSA 433-451…References… I consider the total number of references 9 to be absolutely insufficient. I personally prefer an average of 50 references, 30 references is the lower limit of acceptability for a quality scientific article. More detailed requirements and recommendations are given in the requirements for redesigning the Introduction.
Facultative recommendations:
LNSA 2-3... Quality Identification Method Based on the Difference in Fluidity between Weathered Sand and River Sand... I would like to recommend to the authors a slight modification of the title of reviewed scientific article. Personally, I would use, for example, the following title "Identification fluidity method to determine suitability of weathered and river sand for constructions purposes" (please take it only as an inspiration).
LNSA 77… Zhao Wenkun of CCCC First Harbor Engineering Co…it would be appropriate to give the full name of CCCC abbreviation.
LNSA 111-112…Calculate the natural moisture content of weathered sand:..all equations in a scientific article should be numbered.
LNSA 140… Figure 1. Particle analysis result plot... I recommend the authors to unify the style of graphs throughout the article (color, style and font size, overlay of graphic and text data, ...).
LNSA 167-168.. it is necessary to assign a separate table number to the data presented in tabular form, or to present the data in text form.
LNSA 175...3.3. Test method for loose bulk density of sand: ...I recommend not using a colon at the end of the subsection title, the recommendation applies to the entire article.
LNSA 252... ments:Qn3.0 or less.If...spaces between words are missing.
LNSA 263...â‘¡ Fluidity test...I recommend not to use ring numbers to mark a separate part of the article (subchapter). From my point of view, it is more appropriate to give a separate subchapter or to indicate only highlighted font, the recommendation applies to the entire article.
LNSA 279-281… After the cement mortar mixer is used to mix at a low speed for a set time, the sample is evenly added to the mixture at a speed of 140 ± 5 r/min and 62 ± 5 r/min for the rotation of the mixer shaft at a low speed...I recommend specifying when it will be used 140 ± 5 r/min and when 62 ± 5 r/min are used.
LNSA 395... Conclusion: ...I recommend moving the title of the chapter to the next page and no using the colon.
I consider the scientific potential of the reviewed scientific article to be considerable and, despite the mentioned comments, I recommend its revision. I would like to repeat the emphasis on the requirements for a significant expansion of references, graphic improvements of images and reworking of the abstract, introduction and conclusions.
As already indicated, I highly positively appreciate the authors' effort to exactly quantify the use of conditionally suitable materials for construction purposes, which is in line with my holistic perception in my dominant research area the mixture and structure pavement design. From the aspect of the stated fact and based on a detailed study of the article, in case of incorporation of comments, or relevant justification of their non-incorporation, I am able to process a repeated review within 3 days
Author Response
Thank you for your valuable comments. Here is our reply to your question.

Reviewer 2 Report
The reviewer thinks this manuscript, which deals with fluidity in sand systems, is excellent.
The reviewer appreciates the efforts of the authors.
However, the reviewer requests the following comments to the authors.
The authors should respond to the comments below.
(1) In Section 2, the authors should explain in more detail the background of the sand taken up in this study.
(2) How will the authors incorporate each result obtained in the experiment into the design? Please indicate that perspective in the manuscript.
(3) Related to (2), how do the authors interpret the results of experiments performed on specific sands to generalize?
(4) The authors should provide more comprehensive reviews of similar studies.
(5) The reviewer recommends the authors to fully refer to the sand fluidity evaluation by MPS-CAE simulation.
Author Response

(The authors gave the same response as above.)

Round 2
Reviewer 1 Report
Based on the incorporation of the changes recommended by me, I allow myself to rate the assessed second version of the scientific paper as follows. Reviewed contribution: Identification fluidity method to determine suitability of weathered and river sand for constructions purposes, original title: Quality Identification Method Based on the Difference in Fluidity between Weathered Sand and River Sand. Based on my experience in the assessed issue and subsequent deepening of my knowledge, I am pleased that the submitted 2nd version of the article meets all my essential requirements for a quality scientific article.
I am fully satisfied with the implementation of the required changes and recommendations and I hope that I have contributed a little to improving the quality of the assessed scientific contribution.
In the final version of the article, it would be appropriate to remove the following minor formal deficiencies:
LNSA (Line number of the scientific article) 139…Table 1. Natural water content test.... it is necessary to remove the row numbers from the table.
LNSA 191…Figure 2. Sand apparent density test... the current form of the Figure 2 is acceptable to me.
LNSA 200… there was probably a typo in the article or in equation 3. The correct result is 2551.97 and not 2552.97.
LNSA 356…Table 5. Fluidity test results of Embodiment 1.... wrong table number and you need to move the table title and part of the table to the next page.
LNSA 390… Table 7. Fluidity test results of Embodiment 2...you need to move the table title and part of the table to the next page.
In conclusion, I would like to sincerely congratulate the authors on a very good scientific article and thank the publisher for the opportunity to expand my scientific knowledge in the following field. Using of the innovation method enabling credibility control the quality of river sand, and make the laboratory inspection method standardized management.
Author Response
Thank you for your revision of the article. The reply is as follows.
